

# Effect of pasture and feeding systems on hematological traits of ewes and lambs

Cemil Tölü[1,*], Hülya Hanoğlu Oral[2,*], Fırat Alatürk[3] and Ahmet Gökkuş[4]

[1] Faculty of Agriculture, Department of Animal Science, Çanakkale Onsekiz Mart University, Çanakkale, Turkey
[2] Department of Animal Sciences and Technologies, Faculty of Applied Sciences, Muş Alparslan University, Muş, Turkey
[3] Faculty of Agriculture, Department of Field Crop Science, Çanakkale Onsekiz Mart University, Çanakkale, Turkey
[4] Department of Field Crops/Faculty of Agriculture, Çanakkale Onsekiz Mart University, Canakkale, Center, Turkey
[*] These authors contributed equally to this work.

Corresponding author
Cemil Tölü, cemiltolu@comu.edu.tr

## ABSTRACT

**Context.** Sustainable livestock production depends on efficient pasture management and the continuous monitoring of the health of grazing animals.

**Objectives.** This study investigated the effects of pasture types and sheep production systems on the hematological traits of Karacabey Merino (German Mutton Merino × Kıvırcık) ewes and lambs grazing on different pasture types throughout the year and reared in a semi-intensive system (control group).

**Methods.** In this twenty-six-month study, the hematological characteristics of ewes and lambs grazing on natural pastures and in spring (triticale and oat grass pasture), summer (sorghum Sudangrass and wheat stubble pasture), and autumn (triticale and oat grass pasture) were compared with ewes and lambs reared in a semi-intensive system (no pasture for lambs). A total of 36 ewes (aged 3–4 and 65.2 ± 0.55 kg) and 98 lambs were used. There were 12 ewes in total in each treatment group and four ewes in subgroups. Each subgroup in the pasture was separated by fences. The 12 ewes tagged in the control group were within the unit sheep flock. Blood samples were taken from the jugular vein of ewes and lambs at three to four-week intervals during live weight weighing using 3–4 mL edged tubes and transported in the cold chain to the laboratory for analysis.

**Results.** Pregnant ewes grazing on natural autumn pasture had significantly lower red blood cell (RBC), hemoglobin (HGB), and platelet (PLT) counts ($P < 0.05$). Animals grazing on wheat stubble pasture exhibited higher hematological values compared to those grazing on Sorghum-Sudangrass pasture ($P < 0.05$). Lambs in the control group had lower hemoglobin (HGB) and hematocrit (HCT) levels during the post-weaning period than those in the pasture group ($P < 0.05$).

**Conclusions.** The hematological parameters of pregnant Karacabey Merino ewes grazing on natural pastures during the autumn season, as well as those of lambs raised in a semi-intensive system during the post-weaning period, should be carefully monitored. Additionally, appropriate feed supplementation, along with vitamin and mineral support, should be provided to maintain animal health and physiological balance.

## INTRODUCTION

In today's world, where the effects of global climate change are increasingly evident, the efficient use of natural resources has become critically important. One of the most effective strategies for utilizing natural resources is the grazing of farm animals on pastureland. Pasture-based animal production systems aim to effectively utilize pasture throughout the year (*Celaya et al., 2007*; *Tölü et al., 2017*). Sheep exhibit a distinct utilization of herbaceous plants and demonstrate specialized foraging behaviors compared to other ruminants (*Gupta et al., 2007*; *Barsila et al., 2020*). While the primary objective of grazing-based systems is to utilize natural pastures, artificial pastures established with cereal grasses and legumes reduce the farm's feed costs. In addition, at the point of meat and milk production in pasture-based systems, products can be produced with flavors and tastes that appeal to consumer preferences (*Priolo et al., 2002*; *Hayaloglu et al., 2013*; *Özcan et al., 2014*; *Hanoğlu Oral et al., 2023*).

The main objective of animal production systems is to ensure sustainable animal health alongside an economically efficient and productive process. This is essential for guaranteeing continuous access to healthy animal-derived products for consumers. From a sustainability standpoint, minimizing input costs is critically important for both maintaining animal health and improving product quality (*Silva et al., 2024*). In addition to enhancing reproductive performance, growth, and meat and milk yield, the regular monitoring of animal health parameters is a key factor for production success. The monitoring of hematological traits, an important indicator of animal health, is useful to evaluate physiological changes in animals that remain on pasture throughout the year, to optimize the production efficiency of pasture-based production systems, and to manage and regulate management practices (*Gupta et al., 2007*; *Barsila et al., 2020*). The hematologic values of farm animals are influenced by a multitude of factors, including age, sex, breed, climate, geographical location, season, day length, time of day, nutritional status, living habits of the species, and the current condition of the individual (*Mohammed, Campbell & Youssef, 2014*).

It has been reported that the hematological characteristics of livestock vary depending on animal production systems such as grazing, roughage supplementation, and concentration in the daily diet (*Kochewad et al., 2017*; *Karthik et al., 2021*; *Zaher et al., 2022*). While hemoglobin (Hb), packed cell volume (PCV) and red blood cell (RBC) values of sheep in extensive (with grazing) and semi-intensive production systems (with grazing) have been reported to be lower than in intensive (zero grazing) production systems, white blood cell (WBC) value has been reported to be lower in intensive system than in other systems (*Karthik et al., 2021*). The Hb value is higher in the intensive system and the PCV value is higher in the extensive system in Deccani ewes (*Kochewad et al., 2017*). Furthermore, RBC, mean corpuscular hemoglobin concentration (MCHC), and lymphocyte concentrations

in sheep have been found to vary depending on the addition of concentrate feed to the diet (*Ayele et al., 2017*).

The Karacabey Merino sheep was obtained by crossbreeding the German Mutton Merino sheep with the indigenous Kıvırcık sheep. It is an important genotype (95% and above German Mutton Merino genotypes) regarding lamb production in Türkiye. In this study, we aimed to test the hypothesis that Karacabey Merino ewes and lambs can be produced healthily by utilizing natural and artificial pastures throughout the year. For this purpose, we used hematological traits to test our hypothesis. Karacabey Merino ewes and lambs (up to 4 months of age) were monitored under various grazing systems and a semi-intensive system as a control for about two years. Pasture types were differentiated into spring (triticale and oat grass), summer (sorghum Sudangrass hybrid and wheat stubble), and autumn (triticale and oat grass). In addition, the hematological traits of ewes and lambs reared in year-round pasture systems and semi-intensive animal production systems (pasture and pen conditions) were compared.

## MATERIALS AND METHODS

### Study design and groups

The current study was approved by the Ethics Committee of Çanakkale Onsekiz Mart University (protocol no. 2014/08-01) for studies involving animals. All procedures were conducted in accordance with the "Guiding Principles in the Care and Use of Animals" (Türkiye). All recordings, observations, blood sampling, and lamb slaughter procedures were conducted in accordance with the guidelines approved by the institutional animal ethics committee. A total of five lambs died over two years (three in the pasture groups and two in the control group). No ewe or lamb was euthanized. A total of 58 lambs were slaughtered over two years as required by the animal ethics committee for carcass and meat quality (*Hanoğlu Oral et al., 2023*). At the end of the trial, surviving ewes and lambs were returned to the normal production processing in the unit where the trial was carried out.

This study was carried out over a two-year period at the Sheep Breeding Research Institute located in Bandırma, Balıkesir, in the Marmara Region of Türkiye ($40°17'00''$–$40°20'17''$N; $27°53'37''$–$27°58'25''$E). The study started in April and ended at the end of May after 26 months. In the study, seasonal pastures were established as follows: 1,080 m$^2$ (600 m$^2$ + 480 m$^2$) of triticale pasture and 1,080 m$^2$ (600 m$^2$ + 480 m$^2$) of oat pasture in spring (April–May); 1,200 m$^2$ (600 m$^2$ + 600 m$^2$) of sorghum-Sudangrass (SSG) pasture and 12,000 m$^2$ of wheat stubble pasture in summer (August–September); and 2,600 m$^2$ (2,000 m$^2$ + 600 m$^2$) of triticale pasture and 2,600 m$^2$ (2,000 m$^2$ + 600 m$^2$) of oat pasture in autumn (December–January). The animals were allowed to graze freely on these pastures (Fig. 1). During the remaining periods of the year-spring, autumn, and winter-a 7,200 m$^2$ area of natural pasture was allocated for grazing. For the triticale, oat, and SSG pastures, each area was divided into two subplots to facilitate a rotational grazing system. Grazing commenced in the first plot and shifted to the second once the available forage was depleted, thereby ensuring continuous access to fresh herbage. The size of the plots was determined according to the daily requirement of dry matter at the live weight of the sheep (2.5%).

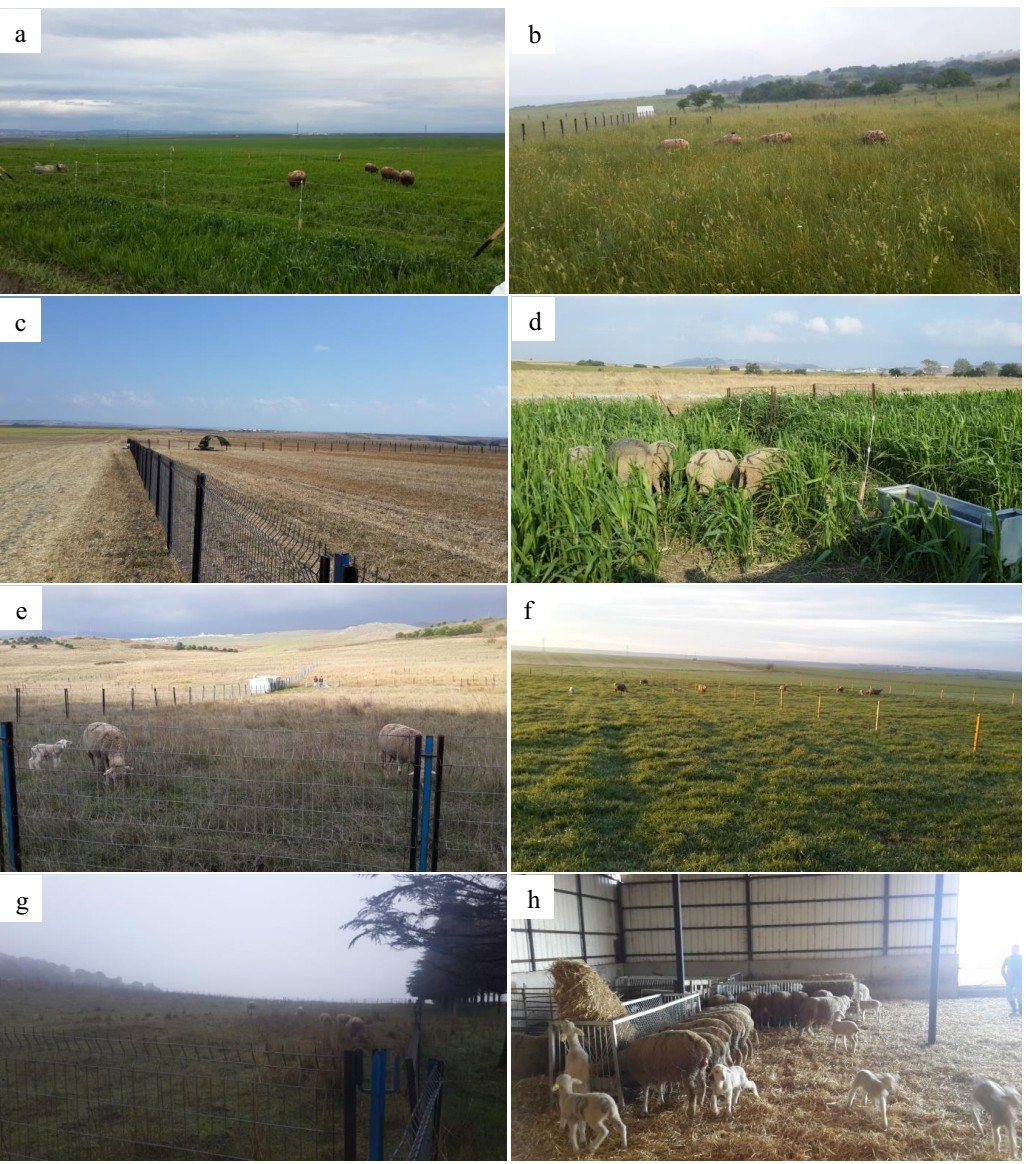

**Figure 1** **The areas of the grazed pasture plots and the control group in the experiment.** (A) Spring cereal grass pasture; (B) Spring nature pasture; (C) wheat stubble pasture; (D) sorghum Sudangrass pasture; (E) autumn nature pasture; (F) autumn cereal grass pasture; (G) winter nature pasture; (H) control group in barn condition.

When grazing on cereal grass and SSG was insufficient, new grazing was fenced off. All pastures plots were similar for soil type, slope and plant density.

During the winter period, including lambing, the ewes were housed in group conditions within a pen for approximately one month. Each pasture parcel consisted of three sub-plots and each sub-plot had four ewes. A total of 24 ewes were included in the pasture plots, while a total of 12 ewes with three subgroups were included in the semi-intensive animal production system conditions routinely practiced at the Research Institute. A total of 36

ewes and 98 lambs of Karacabey Merino were used in the study. Three group replicates were created for each pasture type and control group. All ewes included in the study were between 3 and 4 years of age and had a mean live weight of $65 \pm 0.55$ kg. The ewes were randomly assigned to each of the groups according to their age, body weight, parity, and the number of lambs born at the beginning of the experiment. The ewes were included in the experimental plots after weaning of their offspring in the first year and the experiment was continued with the same ewes for two years. After 10 days of age, the lambs grazed freely with their dams in each plot (Fig. 1). The ewes received no supplementary feed beyond pasture forage. During the summer grazing period, six Karacabey Merino rams—of the same breed as the ewes—were used for mating, with one ram assigned to each pasture plot. The rams were randomly allocated based on age and body weight to ensure balanced distribution among the plots. No additional pasture area was allocated in the pasture plots for rams and lambs. Matings in the control group were performed using the hand-mating method. The control group of ewes was determined to be in oestrus by a teaser ram. They were individually mated with three rams for 10 min.

## Pastures and plant measurements

The grazing plots were surrounded by fences made of five mm iron divided into $15 \times 15$ cm squares. To measure grass yields and the amount of grass consumed, areas protected from grazing were needed. For this, four protected areas of 6.25 m² ($2.5 \times 2.5$ m) were created in each plot. The pasture dry matter yield (DM kg/ha), daily DM intake per ewe (excluding lambs), and nutrient content of pasture plants were determined (*Tölü et al., 2012*; *Hanoğlu Oral et al., 2023*; *Alatürk, 2024*) in the plots throughout the experiment are presented in Table 1.

## Animals and feeding

No supplemental feed was provided to the ewes on pasture, except during the winter period when they were housed in the pen. Throughout the study, the ewes were managed to rely primarily on pasture for their nutritional needs. The routine feeding program of the Research Institute was followed for animals in the control (C) group. The feeding program was designed using the (*National Research Council of the National Academies, 2007*) reference values for mature ewes for the breeding, gestation and lambing periods. The ewes in the control group were grazed by a shepherd for 5–8 h on natural pasture and stubble on the farm, except in Jan-Feb when lambing occurred. Alfalfa hay (16.7% CP and 2,240.2 kcal/kg ME) and vetch hay (13.6% CP; 1,410.4 kcal/kg ME) were used as supplemental roughage for the ewes. The concentrate diet (15.6% CP and 2,457.8 kcal/kg ME) consisted mainly of barley, maize, wheat, sunflower seed meal, marble powder, salt, and a vitamin-mineral mixture. Starting from ten days of age, the lambs were provided with alfalfa and a concentrate feed containing 18% crude protein and 2,700 kcal/kg metabolizable energy, offered *ad libitum*. The concentrated feed was composed of barley, maize, sunflower meal, marble powder, salt, and a vitamin and mineral mixture. The lambs on pasture were provided with concentrate feed *ad libitum* in a semi-enclosed pen located within each pasture parcel, under creep feeding conditions. The lambs grazed freely in

**Table 1  Means of nutrient components, energy values, pasture yields, and daily DM intake values per ewe of pasture types by year during the study period.**

| Pasture | Year | DM, % | CP, % | NDF, % | ADF, % | DOM, % | Ash, % | ME, Mcal/kg | Yield KM, kg/da | Intake DM kg/ewe/day |
|---|---|---|---|---|---|---|---|---|---|---|
| Spring triticale | 1st | 29.07 | 12.68 | 53.64 | 28.06 | 70.34 | 12.67 | 2.58 | 1,226.7 | 1.27 |
| | 2nd | 31.84 | 14.06 | 53.63 | 34.11 | 66.09 | 14.06 | 2.39 | 542.8 | 1.71 |
| Spring oat | 1st | 26.75 | 12.89 | 52.66 | 26.73 | 71.28 | 11.72 | 2.63 | 1,653.8 | 1.81 |
| | 2nd | 26.96 | 15.68 | 50.57 | 33.43 | 66.56 | 15.28 | 2.41 | 356.2 | 1.65 |
| Spring nature-1 | 1st | 48.96 | 6.65 | 61.46 | 21.84 | 72.55 | 9.18 | 2.42 | 342.6 | 1.59 |
| | 2nd | 61.75 | 9.35 | 60.42 | 36.12 | 63.13 | 9.18 | 2.06 | 376.2 | 1.73 |
| Spring nature-2 | 1st | 48.11 | 6.77 | 62.75 | 21.84 | 72.55 | 9.29 | 2.42 | 358.9 | 2.15 |
| | 2nd | 58.45 | 9.64 | 61.21 | 36.02 | 63.19 | 9.36 | 2.07 | 333.6 | 1.86 |
| Wheat stubble | 1st | 87.74 | 6.28 | 63.46 | 34.27 | 65.98 | 13.53 | 2.02 | 644.2 | 2.03 |
| | 2nd | 90.74 | 5.21 | 73.18 | 46.61 | 57.29 | 20.99 | 1.67 | 416.8 | 2.09 |
| Sorghum Su-dangrass | 1st | 30.13 | 11.33 | 61.31 | 31.18 | 68.15 | 10.04 | 2.11 | 1,773.8 | 2.94 |
| | 2nd | 39.47 | 11.12 | 63.07 | 35.06 | 65.42 | 11.44 | 1.99 | 809.5 | 2.09 |
| Autumn Nature-1 | 1st | 64.61 | 5.72 | 65.09 | 23.82 | 71.25 | 10.69 | 2.35 | 391.9 | 2.08 |
| | 2nd | 82.81 | 6.95 | 73.34 | 43.12 | 58.51 | 15.12 | 1.83 | 192.6 | 1.92 |
| Autumn Nature-2 | 1st | 66.65 | 5.72 | 63.69 | 22.66 | 72.01 | 11.12 | 2.39 | 358.1 | 2.62 |
| | 2nd | 85.06 | 6.64 | 74.68 | 45.03 | 57.25 | 14.88 | 1.78 | 182.8 | 1.59 |
| Autumn triti-cale | 1st | 25.00 | 24.73 | 39.25 | 23.42 | 73.62 | 20.84 | 2.73 | 263.1 | 1.16 |
| | 2nd | 28.87 | 19.71 | 44.37 | 22.11 | 74.53 | 15.65 | 2.70 | 281.2 | 1.27 |
| Autumn oat | 1st | 22.90 | 21.87 | 38.06 | 22.86 | 74.00 | 16.78 | 2.70 | 265.0 | 1.32 |
| | 2nd | 25.77 | 21.34 | 44.57 | 24.91 | 72.57 | 14.90 | 2.77 | 265.7 | 1.22 |
| Winter nature-1 | 1st | 58.14 | 11.35 | 61.92 | 36.62 | 62.80 | 16.98 | 2.07 | 261.7 | 1.59 |
| | 2nd | 68.43 | 9.15 | 61.81 | 37.37 | 62.30 | 20.78 | 2.02 | 156.2 | 1.29 |
| Winter nature-2 | 1st | 57.60 | 10.86 | 64.66 | 35.89 | 63.28 | 15.11 | 2.09 | 279.4 | 2.18 |
| | 2nd | 75.30 | 9.24 | 65.01 | 39.32 | 61.02 | 21.16 | 1.97 | 137.0 | 1.56 |

**Notes.**

DM, Dry matter; CP, Crude protein; NDF, Neutral detergent fiber; ADF, Acid detergent fiber; DOM, Digestibility organic matter; ME, Metabolize energy.

the pasture with dams. Lambs in the control group, which were fed in a semi-intensive system, did not have access to pasture and were housed in an indoor pen. The lambs in the control group were weaned at two months of age and fed concentrate feed (16% CP and 2,600 kcal/kg ME). To prevent the lambs from developing urinary stones, 1% ammonium chloride was added to the growing lamb feed. The same feeding program was followed in all years of the experiment.

## Hematological analyses

Blood samples were taken from the vena jugularis at three to four weeks during the live weight weighing of the animals using 3–4 ml edged tubes and transported to the laboratory in the cold chain. Hematological parameters were measured on a fully automated hematology analyzer (Diatron Abacus Junoir Vet 5; Volumetric impedance method, light absorbance for hemoglobin measurement, Diatron MI PLC 1038 Budapest, Hungary) on the same day. Hematological parameters examined were white blood cell (WBC), lymphocyte (LY), neutrophil (NE), lymphocyte to neutrophil ratio (N:L), red blood cell

**Table 2** Least square mean (LSM), standard error of means (SEM), and *P* values of hematological traits of Karacabey Merino ewe in spring according to cereal pasture groups, control group, and years.

| Traits | Triticale | Oat | Control | SEM | 1st year | 2nd year | SEM | Pasture x Year |
|---|---|---|---|---|---|---|---|---|
| | LSM | LSM | LSM | | LSM | LSM | | |
| WBC,$10^9$/L | 8.96 | 8.37 | 8.44 | 0.49 | 9.58$^x$ | 7.59$^y$ | 0.32 | 0.6944 |
| LY, % | 68.55 | 69.10 | 67.86 | 1.27 | 69.87$^x$ | 67.13$^y$ | 0.98 | 0.3506 |
| NE, % | 30.89 | 30.41 | 31.34 | 1.30 | 29.65$^x$ | 32.12$^y$ | 0.99 | 0.3632 |
| N:L | 0.47 | 0.46 | 0.48 | 0.03 | 0.44$^x$ | 0.50$^y$ | 0.02 | 0.2643 |
| RBC,$10^{12}$/L | 9.73 | 9.64 | 9.73 | 0.17 | 10.32$^x$ | 9.07$^y$ | 0.11 | 0.7204 |
| HGB, g/dL | 9.87 | 9.89 | 9.63 | 0.20 | 10.11$^x$ | 9.48$^y$ | 0.13 | 0.2642 |
| HCT, % | 27.08 | 27.55 | 27.15 | 0.56 | 28.46$^x$ | 26.06$^y$ | 0.42 | 0.7501 |
| MCV, fL | 28.60 | 27.91 | 27.88 | 0.45 | 27.66$^x$ | 28.60$^y$ | 0.30 | 0.8130 |
| MCHC, g/dL | 36.55 | 35.85 | 35.56 | 0.32 | 35.48$^x$ | 36.49$^y$ | 0.23 | 0.4086 |
| PLT, $10^9$/L | 343.50 | 343.07 | 392.24 | 28.06 | 363.92 | 355.28 | 19.51 | 0.1436 |

Notes.

WBC, white blood cell; LY, lymphocyte; NE, neutrophil; N:L., lymphocyte to neutrophil ratio; RBC, red blood cell; HGB, hemoglobin; HCT, hematocrit; MCV, mean red blood cell volume; MCHC, mean red blood cell hemoglobin concentration; PLT, platelet (thrombocyte) count.

a, b, Different letters indicate significant differences between pasture groups and x, y, different letters indicate significant differences between years ($P < 0.05$).

(RBC), hemoglobin (HGB), hematocrit (HCT), mean red blood cell volume (MCV), mean red blood cell hemoglobin concentration (MCHC), and platelet count (PLT).

## Statistical analyses

Statistical analyses were performed separately for the ewes and lambs. Analyses were made according to pasture types at six time periods during the year. The control group was included in the study throughout all periods, whereas the sheep (ewes and lambs) grazing on natural pastures constituted the experimental pasture group. For the ewes' analyses, repeated measures analysis of variance was performed including pasture type, year (1st year and, 2nd year), pasture × year interaction, and control day as fixed factors and animal effect as a random factor in the statistical model. In the analyses for the lambs, a repeated analysis of variance was performed including pasture type, year, pasture × year, sex, birth type, control day as fixed factors, and animal effect as a random factor in the statistical model. In the final analyses, sex and birth type were not included in the final analyses as they were not statistically significant. Tukey test was used in *post hoc* analyses. All analyses were performed in (*SAS, 2021*) statistical package program.

## RESULTS

In the spring season, the hematological parameters of the ewes were similar between the cereal grass pasture group and the control group (Table 2). Hematological characteristics were significantly different between years except for PLT ($P < 0.05$). The values for WBC, LY, RBC, HGB, and HCT were higher in the first year than in the second year, whereas the values for NE, N:L, MCV and MCHC were higher in the second year ($P < 0.05$).

Following the spring cereal pasture period, the lymphocyte (LY), neutrophil (NE), and neutrophil-to-lymphocyte ratio (N:L) values of the ewes grazed on natural pastures differed significantly between the pasture and control groups (Table 3). The LY value was higher

**Table 3  Least square mean (LSM), standard error of means (SEM), and *P* values of hematological traits of Karacabey Merino ewe in spring according to natural pasture group, control group, and years.**

| Traits | Pasture | Control | SEM | 1st year | 2nd year | SEM | Pasture x Year |
|---|---|---|---|---|---|---|---|
| | LSM | LSM | | LSM | LSM | | |
| WBC,$10^9$/L | 10.17 | 10.03 | 0.36 | 11.71$^x$ | 10.82$^y$ | 0.31 | 0.0192 |
| LY, % | 66.52$^a$ | 63.03$^b$ | 1.15 | 66.51$^x$ | 71.04$^y$ | 0.94 | 0.0086 |
| NE, % | 32.98$^a$ | 36.46$^b$ | 1.16 | 32.98$^x$ | 28.47$^y$ | 0.98 | 0.0088 |
| N:L | 0.52$^a$ | 0.62$^b$ | 0.03 | 0.52$^x$ | 0.41$^y$ | 0.02 | 0.0222 |
| RBC,$10^{12}$/L | 9.27 | 9.40 | 0.17 | 9.20 | 8.96 | 0.14 | 0.0002 |
| HGB, g/dL | 9.37 | 9.52 | 0.16 | 9.20$^x$ | 8.96$^y$ | 0.12 | 0.0656 |
| HCT, % | 26.29 | 26.55 | 0.41 | 25.91 | 25.27 | 0.34 | <0.0001 |
| MCV, fL | 28.54 | 28.10 | 0.34 | 28.50$^x$ | 27.91$^y$ | 0.26 | 0.1085 |
| MCHC, g/dL | 35.68 | 35.89 | 0.24 | 35.58 | 35.51 | 0.22 | 0.1738 |
| PLT, $10^9$/L | 286.89 | 348.89 | 17.29 | 310.92 | 301.68 | 14.62 | 0.7245 |

**Notes.**

WBC, white blood cell; LY, lymphocyte; NE, neutrophil; N:L:, lymphocyte to neutrophil ratio; RBC, red blood cell; HGB, hemoglobin; HCT, hematocrit; MCV, mean red blood cell volume; MCHC, mean red blood cell hemoglobin concentration; PLT, platelet (thrombocyte) count. a, b, Different letters indicate significant differences between pasture groups and x, y, different letters indicate significant differences between years ($P < 0.05$).

in the pasture group compared to the control group, while the NE and the N:L ratio were higher in the ewes in the control group ($P < 0.05$). While the values for WBC, NE, N:L, and MCV were higher the first year than in the second year, the value for LY was higher in the second year ($P < 0.05$).

LY, NE, N:L, RBC, HGB, MCV, MCHC and PLT values of of the hematological characteristics of Karacabey Merino ewes grazing on wheat stubble, sorghum Sudangrass (SSG) pastures, and in the control group differed significantly among the groups (Table 4). LY, NE, and N:L values were significantly different between the SSG group and the control group, and RBC and HGB values were significantly different between the stubble group and the SSG group ($P < 0.05$). MCV, MCHC and PLT values of the ewes in the SSG pasture group were significantly different from those in the stubble and control groups ($P < 0.05$). WBC, LY, NE, N:L, HGB, HCT, MCV, and PLT values changed according to years ($P < 0.05$). WBC, NE, N:L, HGB, and HCT were higher in the first year, whereas LY, MCV, and PLT were higher in the second year ($P < 0.05$).

The hematological traits of the ewes grazing on natural pastures during the autumn season and the ewes in the control group were similar (Table 5). WBC, LY, NE, RBC, HCT, MCHC, and PLT values differed between years ($P < 0.05$). WBC, NE, RBC, HCT, and PLT were higher in the first year, whereas LY and MCHC were higher in the second year ($P < 0.05$).

The hematological values of the ewes on the cereal grass pastures established in the autumn were similar to the control group (Table 6). WBC, RBC, HGB, HCT and MCHC values showed differences between years ($P < 0.05$). While WBC, RBC and HCT values were higher in the first year, HGB and MCHC values were higher in the second year ($P < 0.05$).

**Table 4** Least square mean (LSM), standard error of means (SEM), and *P* values of hematological traits of Karacabey Merino ewe in summer according to pasture groups, control group, and years.

| Traits | Stubble | SSG | Control | SEM | 1st year | 2nd year | SEM | Pasture x Year |
|---|---|---|---|---|---|---|---|---|
| | LSM | LSM | LSM | | LSM | LSM | | |
| WBC, $10^9$/L | 9.74 | 9.56 | 9.92 | 0.49 | 10.36[x] | 9.12[y] | 0.35 | 0.0040 |
| LY, % | 69.35[ab] | 66.24[a] | 71.89[b] | 1.61 | 65.95[x] | 72.38[y] | 1.17 | 0.0300 |
| NE, % | 30.14[ab] | 33.25[a] | 27.62[b] | 1.61 | 33.55[x] | 27.13[y] | 1.17 | 0.0298 |
| N:L | 0.47[ab] | 0.55[a] | 0.41[b] | 0.03 | 0.56[x] | 0.39[y] | 0.02 | 0.0576 |
| RBC, $10^{12}$/L | 8.97[a] | 8.26[b] | 8.79[a] | 0.17 | 8.70 | 8.64 | 0.12 | 0.0463 |
| HGB, g/dL | 9.24[a] | 8.50[b] | 8.94[ab] | 0.18 | 9.03[x] | 8.76[y] | 0.13 | 0.0015 |
| HCT, % | 26.04 | 25.22 | 25.49 | 0.47 | 25.92[x] | 25.24[y] | 0.33 | 0.0094 |
| MCV, fL | 29.22[a] | 30.50[b] | 28.83[a] | 0.45 | 29.93[x] | 29.11[y] | 0.30 | 0.8443 |
| MCHC, g/dL | 35.46[a] | 33.86[b] | 34.88[a] | 0.34 | 34.79 | 34.67 | 0.24 | 0.2291 |
| PLT, $10^9$/L | 344.53[a] | 272.59[b] | 356.42[a] | 19.60 | 291.93[x] | 357.09[y] | 14.59 | 0.0004 |

Notes.
SSG, Sorghum-Sudangrass; WBC, white blood cell; LY, lymphocyte; NE, neutrophil; N:L, lymphocyte to neutrophil ratio; RBC, red blood cell; HGB, hemoglobin; HCT, hematocrit; MCV, mean red blood cell volume; MCHC, mean red blood cell hemoglobin concentration; PLT, platelet (thrombocyte) count.
a, b, Different letters indicate significant differences between pasture groups and x, y, different letters indicate significant differences between years (*P* < 0.05).

**Table 5** Least square mean (LSM), standard error of means (SEM), and *P* values of hematological traits of Karacabey Merino ewes in autumn according to natural pasture group, control group, and years.

| Traits | Pasture | Control | SEM | 1st year | 2nd year | SEM | Pasture x Year |
|---|---|---|---|---|---|---|---|
| | LSM | LSM | | LSM | LSM | | |
| WBC, $10^9$/L | 8.62 | 9.23 | 0.38 | 9.64[x] | 8.21[y] | 0.33 | 0.0014 |
| LY, % | 70.28 | 71.49 | 1.03 | 68.76[x] | 73.01[y] | 0.91 | 0.0246 |
| NE, % | 29.23 | 28.02 | 1.02 | 33.02[x] | 28.48[y] | 0.90 | 0.0247 |
| N:L | 0.42 | 0.41 | 0.01 | 0.47[x] | 0.37[y] | 0.01 | 0.0886 |
| RBC, $10^{12}$/L | 7.87[a] | 8.54[b] | 0.15 | 8.98[x] | 7.53[y] | 0.11 | 0.5564 |
| HGB, g/dL | 9.49 | 9.81 | 0.14 | 9.73 | 9.56 | 0.11 | 0.0410 |
| HCT, % | 26.71[a] | 28.02[b] | 0.43 | 27.55 | 27.18 | 0.34 | 0.9469 |
| MCV, fL | 30.56 | 30.27 | 0.35 | 30.38 | 30.44 | 0.30 | 0.3319 |
| MCHC, g/dL | 35.45 | 35.33 | 0.25 | 35.51 | 35.27 | 0.21 | 0.7971 |
| PLT, $10^9$/L | 276.75[a] | 354.92[b] | 17.65 | 214.12[x] | 407.56[y] | 16.28 | 0.7226 |

Notes.
WBC, white blood cell; LY, lymphocyte; NE, neutrophil; N:L, lymphocyte to neutrophil ratio; RBC, red blood cell; HGB, hemoglobin; HCT, hematocrit; MCV, mean red blood cell volume; MCHC, mean red blood cell hemoglobin concentration; PLT, platelet (thrombocyte) count.
a, b, Different letters indicate significant differences between pasture groups and x, y, different letters indicate significant differences between years (*P* < 0.05).

The LY, NE, and N:L values of the ewes grazing on natural pasture in winter and the ewes in the control group were significant (Table 7). WBC, HGB, and MCHC values differed according to years (*P* < 0.05). While the WBC value was higher in the first year, HGB and MCHC values were higher in the second year (*P* < 0.05).

The scores for LY, NE, N:L and PLT of the lambs on the cereal grass pasture groups were significantly different from the lambs in the control group (Table 8). The hematological

**Table 6** Least square mean (LSM), standard error of means (SEM) and *P* values of hematological traits of Karacabey Merino ewe in autumn according to cereal pasture groups, control group, and years.

| Traits | Triticale | Oat | Control | SEM | 1st year | 2nd year | SEM | Pasture x Year |
|---|---|---|---|---|---|---|---|---|
| | LSM | LSM | LSM | | LSM | LSM | | |
| WBC,$10^9$/L | 9.78 | 9.89 | 10.28 | 0.57 | 10.96$^x$ | 9.01$^y$ | 0.40 | 0.0005 |
| LY, % | 71.22 | 69.65 | 71.49 | 1.71 | 71.67 | 69.60 | 1.34 | 0.1218 |
| NE, % | 28.29 | 29.86 | 28.01 | 1.71 | 27.84 | 29.60 | 1.34 | 0.1234 |
| N:L | 0.41 | 0.45 | 0.41 | 0.03 | 0.41 | 0.44 | 0.02 | 0.1006 |
| RBC,$10^{12}$/L | 7.04 | 7.26 | 7.63 | 0.21 | 8.54$^x$ | 6.08$^y$ | 0.18 | 0.0435 |
| HGB, g/dL | 9.28 | 9.44 | 9.68 | 0.16 | 9.21$^x$ | 9.72$^y$ | 0.14 | 0.0145 |
| HCT, % | 25.66 | 26.30 | 26.90 | 0.62 | 25.93$^x$ | 26.65$^y$ | 0.47 | 0.0010 |
| MCV, fL | 30.30 | 30.21 | 29.81 | 0.46 | 29.86 | 30.36 | 0.30 | 0.0991 |
| MCHC, g/dL | 36.52 | 36.16 | 36.10 | 0.37 | 35.98 | 36.55 | 0.27 | 0.1332 |
| PLT, $10^9$/L | 285.96 | 301.20 | 308.46 | 27.41 | 287.59 | 309.48 | 19.18 | 0.0195 |

Notes.
WBC, white blood cell; LY, lymphocyte; NE, neutrophil; N:L., lymphocyte to neutrophil ratio; RBC, red blood cell; HGB, hemoglobin; HCT, hematocrit; MCV, mean red blood cell volume; MCHC, mean red blood cell hemoglobin concentration; PLT, platelet (thrombocyte) count. a, b, Different letters indicate significant differences between pasture groups and x, y, different letters indicate significant differences between years (*P* < 0.05).

**Table 7** Least square mean (LSM), standard error of means (SEM), and *P* values of hematological traits of Karacabey Merino ewes in winter according to natural pasture group, control group, and years.

| Traits | Pasture | Control | SEM | 1st year | 2nd year | SEM | Pasture x Year |
|---|---|---|---|---|---|---|---|
| | LSM | LSM | | LSM | LSM | | |
| WBC,$10^9$/L | 10.73 | 9.61 | 0.38 | 10.86$^x$ | 9.47$^y$ | 0.33 | 0.0509 |
| LY, % | 72.13$^a$ | 75.28$^b$ | 1.13 | 73.43 | 73.98 | 0.94 | 0.4238 |
| NE, % | 27.38$^a$ | 24.01$^b$ | 1.12 | 26.07 | 25.32 | 0.93 | 0.4529 |
| N:L | 0.39$^a$ | 0.33$^b$ | 0.02 | 0.36 | 0.35 | 0.01 | 0.5870 |
| RBC,$10^{12}$/L | 8.17 | 8.26 | 0.14 | 8.28 | 8.14 | 0.12 | 0.2226 |
| HGB, g/dL | 8.74 | 8.61 | 0.13 | 8.53$^x$ | 8.81$^y$ | 0.11 | 0.9943 |
| HCT, % | 23.61 | 23.60 | 0.36 | 23.82 | 23.37 | 0.31 | 0.5941 |
| MCV, fL | 29.01 | 28.56 | 0.43 | 28.83 | 28.74 | 0.33 | 0.2394 |
| MCHC, g/dL | 37.03 | 36.55 | 0.23 | 35.80$^x$ | 37.77$^y$ | 0.21 | 0.3393 |
| PLT, $10^9$/L | 273.02 | 292.75 | 17.55 | 288.21 | 277.57 | 15.05 | 0.7493 |

Notes.
WBC, white blood cell; LY, lymphocyte; NE, neutrophil; N:L., lymphocyte to neutrophil ratio; RBC, red blood cell; HGB, hemoglobin; HCT, hematocrit; MCV, mean red blood cell volume; MCHC, mean red blood cell hemoglobin concentration; PLT, platelet (thrombocyte) count. a, b, Different letters indicate significant differences between pasture groups and x, y, different letters indicate significant differences between years (*P* < 0.05).

measurements were similar between the pasture groups (*P* > 0.05). The values for N:L, HCT and PLT were higher in the first year than in the second year (*P* < 0.05).

WBC, HGB and HCT values for the lambs grazing on natural pasture were significantly higher than for the lambs in the control group (Table 9). The PLT value was higher in the control group than in the natural pasture group (*P* < 0.05). Hematological characteristics except WBC, HGB and PLT differed between years. While NE, N:L, RBC and HCT values

**Table 8** Least square mean (LSM), standard error of means (SEM) and *P* values of hematological traits of Karacabey Merino sheep lambs in spring according to cereal pasture groups, control group, and years.

| Traits | Triticale | Oat | Control | SEM | 1st year | 2nd year | SEM | Pasture x Year |
|---|---|---|---|---|---|---|---|---|
| | LSM | LSM | LSM | | LSM | LSM | | |
| WBC,$10^9$/L | 12.32 | 11.89 | 10.85 | 0.53 | 11.76 | 11.61 | 0.43 | 0.9431 |
| LY, % | 75.06[a] | 75.73[a] | 78.65[b] | 1.18 | 75.19 | 77.77 | 0.95 | 0.1313 |
| NE, % | 24.36[a] | 23.76[a] | 20.85[b] | 1.17 | 24.26 | 21.73 | 0.94 | 0.1388 |
| N:L | 0.36[a] | 0.33[a] | 0.27[b] | 0.22 | 0.35[x] | 0.29[y] | 0.02 | 0.1108 |
| RBC,$10^{12}$/L | 11.09 | 10.76 | 11.03 | 0.18 | 11.09 | 10.83 | 0.14 | 0.4776 |
| HGB, g/dL | 10.51 | 10.30 | 10.59 | 0.14 | 10.53 | 10.40 | 0.12 | 0.0014 |
| HCT, % | 28.35 | 28.31 | 28.47 | 0.47 | 29.11[x] | 24.64[y] | 0.40 | 0.0025 |
| MCV, fL | 25.54 | 26.26 | 26.12 | 0.36 | 26.36 | 25.59 | 0.30 | 0.0387 |
| MCHC, g/dL | 37.30 | 36.55 | 37.59 | 0.55 | 36.65 | 37.65 | 0.48 | 0.4640 |
| PLT, $10^9$/L | 524.84[a] | 505.72[a] | 602.80[b] | 26.30 | 620.90[x] | 468.00[y] | 21.65 | 0.3273 |

Notes.

WBC, white blood cell; LY, lymphocyte; NE, neutrophil; N:L., lymphocyte to neutrophil ratio; RBC, red blood cell; HGB, hemoglobin; HCT, hematocrit; MCV, mean red blood cell volume; MCHC, mean red blood cell hemoglobin concentration; PLT, platelet (thrombocyte) count. a, b, Different letters indicate significant differences between pasture groups and x, y, different letters indicate significant differences between years ($P < 0.05$).

**Table 9** Least square mean (LSM), standard error of means (SEM), and *P* values of hematological traits of Karacabey Merino sheep lambs in spring according to natural pasture group, control group, and years.

| Traits | Pasture LSM | Control LSM | SEM | 1st year LSM | 2nd year LSM | SEM | Pasture x Year |
|---|---|---|---|---|---|---|---|
| WBC,$10^9$/L | 11.64[a] | 10.54[b] | 0.44 | 11.32 | 10.86 | 0.43 | 0.1228 |
| LY, % | 80.63 | 79.84 | 0.76 | 78.76[x] | 81.71[y] | 0.77 | 0.9275 |
| NE, % | 18.94 | 19.65 | 0.76 | 20.81[x] | 17.79[y] | 0.77 | 0.8651 |
| N:L | 0.24 | 0.26 | 0.01 | 0.28[x] | 0.22[y] | 0.01 | 0.9741 |
| RBC,$10^{12}$/L | 10.02 | 10.06 | 0.13 | 10.47[x] | 9.61[y] | 0.12 | 0.5545 |
| HGB, g/dL | 10.18[a] | 9.77[b] | 0.11 | 9.98 | 10.01 | 0.11 | 0.0801 |
| HCT, % | 26.81[a] | 25.84[b] | 0.35 | 26.89[x] | 25.76[y] | 0.36 | 0.1122 |
| MCV, fL | 26.68 | 25.95 | 0.27 | 25.46[x] | 27.17[y] | 0.26 | 0.0681 |
| MCHC, g/dL | 38.16 | 38.13 | 0.44 | 38.18[x] | 39.10[y] | 0.43 | 0.6232 |
| PLT, $10^9$/L | 505.99[a] | 571.60[b] | 17.54 | 535.84 | 541.76 | 17.39 | 0.6769 |

Notes.

WBC, white blood cell; LY, lymphocyte; NE, neutrophil; N:L., lymphocyte to neutrophil ratio; RBC, red blood cell; HGB, hemoglobin; HCT, hematocrit; MCV, mean red blood cell volume; MCHC, mean red blood cell hemoglobin concentration; PLT, platelet (thrombocyte) count. a, b, Different letters indicate significant differences between pasture groups and x, y, different letters indicate significant differences between years ($P < 0.05$).

were higher in the first year compared to the second year, LY, MCV and MCHC values were higher in the second year ($P < 0.05$).

## DISCUSSION

The Karacabey Merino ewes and lambs in both the pasture groups, which grazed on different forage types over a two-year period, and the control group managed under a semi-intensive production system, did not exhibit any health issues. Moreover, no

clinical abnormalities were detected in the monitored hematological parameters. The hematological values recorded across the six pasture types and periods remained within the reference ranges previously reported for sheep (*Jain, 1993*; *Kramer, 2000*; *Meyer & Harvey, 2004*; *Žura Žaja et al., 2019*; *Newcomer et al., 2021*). Commonly reported hematological reference values for sheep are $4–12 \times 10^9$/L for WBC, $9–15 \times 10^9$/L for RBC, 9–15 g/dL for HGB, 27–45% for HCT, 28–40 fL for MCV, 31–34 g/dL for MCHC and $200–1,100 \times 10^9$/L for PLT (*Kramer, 2000*; *Žura Žaja et al., 2019*; *Newcomer et al., 2021*). However, slightly broader reference ranges have also been reported as $8–16 \times 10^9$/L for RBC, 8–16 g/dL for HGB, 24–50% for HCT, 23–48 fL for MCV, and 31–38 g/dL for MCHC (*Jain, 1993*; *Meyer & Harvey, 2004*). The reference values of the ewes in the study showed that RBC values were below $8.0 \times 10^9$/L when the ewes were in autumn pasture (Table 6), and HCT values were below 24% when the sheep were in winter pasture (Table 7). These levels were considered indicative of anemia in sheep. However, no significant differences in RBC and HCT values were observed between pasture types and control groups during autumn and winter pastures. It was determined that red blood cell measurements decreased with pregnancy in Dorper sheep (*Santarosa et al., 2022*). Although HCT value below 24% is considered as anemia for sheep (*Polizopoulou, 2010*), it is noteworthy that HGB and HCT values were close to the clinical limit values in the present study. The nutrient content of pastures should also be considered during the gestation period. However, it seems difficult to draw a clear conclusion about the effect of pasture quality with the data available in this study. It can be stated that due to increased iron requirements and deficiencies of B vitamins (B6, B9, B12), the risk of anemia increases during pregnancy. In this study, which is part of the project, no statistical difference was observed in Fe measurements in pasture plants. However, it would be better to use pasture, feed, and animal measurements to assess the adequacy of Fe and B vitamins in sheep during this period. The fact that the micronutrient profiles of the blood were not analyzed in the study limits a more definitive discussion of the results of the study.

The greatest variation in hematological data between pasture types and the control group in terms of hematological data was recorded during the period when stubble and SSG pastures were used (Table 4). The ewes grazing on wheat stubble had significantly higher RBC, HGB, and PLT values than the ewes grazing on the SSG pasture. The positive effect of wheat stubble on these hematological values is probably due to the cereal residues it contains, in particular to its fiber or lignin content. Although not statistically significant, the Fe mineral analyses of the pasture plants in the project showed a higher Fe content in the stubble pasture than in the SSG pasture. Therefore, wheat stubble seems to increase these blood values with its rich vitamin and mineral content. In addition, the RBC values measured in the control ewes were like those measured in the stubble-grazed ewes. It is possible that the cereal supplementation given to the control group ewes during this period had an effect. It was found that feeding sheep a mixture of grass-clover and additional barley, alongside grazing, increases Hb and MCH parameters, while supplementing this diet with concentrates significantly increased HCT, MCV, and MCHC values (*Tuncer, 2021*). In another study in sheep, MCH and lymphocyte values were significantly higher in the group with a higher concentrate feed ratio (*Ayele et al., 2017*). In WAD sheep, while

most of the hematological values monitored in the groups in which *A. brasiliana* and *Guinea* grass were added proportionally to the daily ration were similar between groups, Hb and PCV increased in parallel with the rate of addition to the ration (*Mako, Ikusika & Akinmoladun, 2021*). *Awawdeh, Dager & Obeidat (2020)* found no significant difference in hematological characteristics between groups whose daily diets were differentiated in Awassi sheep lambs.

The proportions of alfalfa and barley in the ration were modified, resulting in the observation that sheep fed a barley-dominant ration had higher HCT values and lower MCV and MCHC values (*Ragen et al., 2021*). In our study, those ewes grazing on stubble had higher values for RBC, HGB, and MCHC and lower values for MCV than the ewes grazing on SSG. In addition, the N:L ratio, which can be used as an indicator of stress (*Polizopoulou, 2010*; *Sluistryowati et al., 2022*), was highest in sheep grazing on SSG pasture and this value was higher than in the stubble and control groups. In the behavioral observations made in the SSG pasture, sheep were stressed due to fear of plant height while grazing since there were four heads in each pasture parcel (*Tölü et al., 2018*). However, considering the range of 0.30−0.50 for the ratio N:L (*Sluistryowati et al., 2022*) and the values of 1:2 or 0.5 (*Polizopoulou, 2010*), it can be stated that the ewes in the study were within these ranges. The N:L value for the ewes only on the SSG pasture was measured as 0.55 (Table 4). It can also be noticed that the hematological values of the blood of the control group during this period were better than those of the ewes grazing on the SSG pasture. The absence of a behavioral stress assessment (*e.g.*, cortisol levels, fecal indicators) in the study limits the ability to draw more definitive conclusions from the results.

During the autumn period of natural pasture use, which coincides with the onset of pregnancy in ewes, there was a notable decline in RBC, HCT and PLT values in the ewes on pasture (Table 5). RBC and HCT values were slightly below the reference values (*Jain, 1993*; *Kramer, 2000*; *Meyer & Harvey, 2004*; *Žura Žaja et al., 2019*; *Newcomer et al., 2021*). It was found that grazing sheep in the extensive and semi-intensive production systems had significantly lower HGB, PCV, and RBC values than those in the non-grazing intensive production system (*Karthik et al., 2021*). Although the low levels of RBC and HCT may be attributed to physiological changes associated with pregnancy (*Santarosa et al., 2022*), it is noteworthy that the ewes in the control group were also pregnant, just like those on pasture. It can thus be posited that deficiencies in red blood cell production are present in ewes that graze on pasture, rendering this type of pasture inadequate during the gestational period. Daily dry matter intake per ewe was also found to be low on this pasture, although it varied from year to year, particularly in the second year (Table 1). Furthermore, the values of CP and DOM in these pastures remained at low levels during this period. RBC production was likely reduced on the natural autumn pasture, where the levels of DM, CP, and DOM, as well as DMI, varied due to the presence of dry vegetation and autumn rains. For the Karacabey Merino sheep grazing year-round, this situation needs to be considered to ensure a healthy pregnancy and maternal health. However, when the hematological data and the nutrient contents of the pasture types were taken into consideration, the pasture × year interactions were found to be statistically significant. Consequently, the issue should be dealt with the utmost caution. In fact, the ewes did not experience any health problems

related to their health and pregnancy during the study period. The fact that the PLT values of the ewes in the pasture group were lower than those of the sheep in the control group can be interpreted as a negative effect of the pasture during this period. It should however be noted that the measured PLT value is not less than 100,000 per microliter, which is considered a critical value (*Polizopoulou, 2010*).

The hematological values of the lambs that were kept with their mothers for approximately four months showed notable differences depending on the pasture type and in comparison to the control group. While LY, NE, and N:L values in the control group differed from those in other groups grazing on spring cereal grass pasture (Table 8), the lambs in the control group had lower HGB and HCT values and higher PLT values than those grazing on spring natural pasture (Table 9). It can be posited that the presence of lambs on pasture exerts a beneficial influence on red blood cell parameters. However, the HGB and HCT values of the lambs kept in pens under semi-intensive conditions were observed to fall below the established reference intervals-HGB: 105–137 g/L and HCT: 0.28–0.39 L/L-as reported in the literature (*Lepherd et al., 2009*; *Awawdeh, Dager & Obeidat, 2020*), indicating a risk of anemia. In this context, interventions may be considered to support RBC production in lambs fed in pens at the age of 3–4 months prior to slaughter. For this purpose, dietary Fe supplementation or providing partial access to pasture can be proposed as potential strategies. *Bozdoğan et al. (2003)* reported that sheep aged between five and seven months that had grazed on pasture exhibited significantly higher hemoglobin and erythrocyte levels compared to male Tuj sheep aged between two and three months that had been raised in confinement. Although PLT values differed among lamb groups, all values remained within the reference limits (*Lepherd et al., 2009*). Similarly, the LY:NE ratio in both lambs and ewes showed significant variation depending on the type of pasture and compared to the control group. These LY:NE ratios, although within normal limits (*Kramer, 2000*), may be influenced by different physiological mechanisms in lambs and ewes, possibly due to thermal stress (*Polizopoulou, 2010*; *Karthik et al., 2021*). Therefore, with the current dataset, it is not possible to make a more detailed assessment of LY and NE values for each pasture type. Additionally, WBC levels were higher in the lambs that grazed on natural pastures only during spring, compared to those in the control group (Table 9). Higher WBC values were measured in sheep raised under extensive production systems compared to those in other production systems, particularly during the summer period (*Karthik et al., 2021*). Additional evidence is required to definitively conclude that the lambs experienced heat stress on the natural pasture during the spring period. However, the WBC value measured in those lambs on pasture ($11.64 \times 10^9$/L) remained within the established reference range (*Lepherd et al., 2009*).

## CONCLUSIONS

The hematological parameters (HGB, HCT, and PLT) of Karacabey Merino ewes grazing on different types of pastures throughout the year remained below normal limits during the gestation period. Grazing was found to have a beneficial effect on RBC parameters in lambs, whereas restricted housing conditions following weaning adversely impacted

lamb hematology. These findings highlight the importance of closely monitoring the hematological status of grazing ewes, particularly during pregnancy, as well as that of lambs reared under confined conditions post-weaning. Notably, the stubble pasture utilized in the study had a positive effect on RBC production.

## ACKNOWLEDGEMENTS

The authors would like to express their sincere gratitude to the Bandırma Sheep Breeding Research Institute for its support. Special thanks are also extended to Prof. Dr. Altıngül Özaslan Parlak for her valuable contributions throughout the course of the study.

### Funding

This study was funded by The Scientific and Technological Research Council of Türkiye (Project number: 214O233). The funders had no role in study design, data collection and analysis, decision to publish, or preparation of the manuscript.

### Grant Disclosures

The following grant information was disclosed by the authors:
The Scientific and Technological Research Council of Türkiye: 214O233.

### Competing Interests

The authors declare there are no competing interests.

### Author Contributions

- Cemil Tölü conceived and designed the experiments, performed the experiments, analyzed the data, prepared figures and/or tables, authored or reviewed drafts of the article, and approved the final draft.
- Hülya Hanoğlu Oral conceived and designed the experiments, performed the experiments, analyzed the data, authored or reviewed drafts of the article, and approved the final draft.
- Fırat Alatürk performed the experiments, authored or reviewed drafts of the article, and approved the final draft.
- Ahmet Gökkuş conceived and designed the experiments, performed the experiments, authored or reviewed drafts of the article, and approved the final draft.

### Animal Ethics

The following information was supplied relating to ethical approvals (i.e., approving body and any reference numbers):

The current study was approved by the Ethics Committee of Çanakkale Onsekiz Mart University (protocol no. 2014/08-01) for studies involving animals. All procedures were conducted in accordance with the ''Guiding Principles in the Care and Use of Animals'' (Türkiye).

## Data Availability

Raw data is available in the Supplemental Files.

## Supplemental Information

Supplemental information for this article can be found online at http://dx.doi.org/10.7717/peerj.19846#supplemental-information.

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
