# Peer review of "Effect of pasture and feeding systems on hematological traits of ewes and lambs"

_PeerJ, doi:10.7717/peerj.19846_

## Round 0.1 · original submission · Major Revisions

I have received comments from two Reviewers. Please see below and on the attached pdf one reviewer has provided. Your manuscript requires major revisions before it can be re-considered for publication.

Both reviewers agree that your study is interesting, however raise significant concerns with the data presentation, interpretation and clarity. Please pay particular attention to these points during the revision process.

When revising your manuscript, please consider all issues mentioned in the Reviewers' comments carefully: please outline every change made in response to their comments and provide suitable rebuttals for any comments not addressed.

Please note that given the extent of the suggested changes, your revised submission will need to be re-reviewed, and that an invitation to revise your manuscript does not represent a commitment to eventual publication.

**Language Note:** The review process has identified that the English language must be improved. PeerJ can provide language editing services - please contact us at [email protected] for pricing (be sure to provide your manuscript number and title). Alternatively, you should make your own arrangements to improve the language quality and provide details in your response letter. – PeerJ Staff

·

Basic reporting

Overall impression: The results have been drawn from a long-term experiment and are interesting. However, data presentation is ambiguous and vague for readers.

Experimental design

There are some points suggested to improve this section. Some clarifications are needed for the sheep selection.

Validity of the findings

The report might be valid when modified as suggested.

Additional comments

Review report:
Overall impression: The results have been drawn from a long-term experiment and are interesting. However, data presentation is ambiguous and vague for readers. Here are some of the suggestions for improving the manuscript quality:
Abstract section
• Context: check grammar (is/are) in the first opening sentence.
• Objectives: check grammar (pasture type/pasture types)? Sheep/animal production?
• Karacabey Merino is shown as a crossbred, so give the original information indicating male/female crossing of German one and native Turkish one, i.e., (zzzz x bbbb).
• Methodology: The section does not describe what the control group used and how the sheep were allocated for the experiment. i.e., similarity in body weight, lamb weight. The criteria of experimental sheep selection are avoided in the manuscript.
• Results: The section is not concrete and needs to be rewritten.
• Conclusion: Do you mean for an extra feed supplement or anything else to be administered for pregnant sheep? That must be clear.

Keywords: Use instead the meaningful keywords by replacing older ones.

INTRODUCTION:
• The section lacks a clear statement of why this experiment is done. There are unclear and some necessary sentences (line 54- first sentence, line 75,
• The very first paragraph is confusing, whether it is a part of the introduction or it is a part of the methodology??
• This section is advised to be rewritten, integrating the clear hypotheses/objectives.

MATERIALS AND METHODS:
1. Line 101-103: Provide a clear statement if there is a death of ewes/lambs or not. It is unclear whether it is important for the highlight of the sentence.
2. Is it enough to feed the sheep for the grazing duration from the stated pasture area? State how you allocate the grazing plot size maximum number of sheep and lambs. Did you use the stocking density/stocking rate principles?
3. Line 119-122: Does the same protected/fenced pasture unit provide enough for rams used for grazing together? State concisely how you allocated them for co-grazing with ewes? Are all the rams per unit of the same age/similar weight/ breed??
In principle, the pasture loss can be expected during grazing by dung, urination, and treading damages too/ How did you prevent it, or did you add extra consideration?
4. Line 125: the correct cross symbol for 15x15 cm. Follow the journal guidelines.
5. Line 135: If it is an NRC standard, include at least the CP and TDN considerations.
6. 6. Line 155: For each instrument used, provide the information (manufacturer, trademark, city, country)
7. It is suggested to use the shortest analysis possible. Use three-way analyses to produce fewer tables instead of many fragmented tables. E.g., Pasture type, season, year??? may fit good than the current presentation. That adheres compacted information from a single table.
8. If the interaction effect is non-significant, there is no meaning in presenting the single effect with a mean difference, i.e., year1/year2.

DISCUSSION
• Line 226-229: Is this your objective to highlight in the discussion section? If it is, then revise the introduction section with it.
• The pasture condition (wheat stubble may have more fibre/lignin) in winter might cause greater variation. See at least the forage values to discuss.
• 289-290: Are the authors sure that the decline in blood values is only due to pasture quantity? Pasture quality may be the cause; discuss it.
• Line 296-300: Do you mean resistant to changes in pasture quality?
• 312-313: Add some references for that if it is in your case too.
• Follow a three-way interaction for most of the parameters and revise the section well.

CONCLUSION: A maybe statement would not be a good conclusion. Revise as such from a three-way analysis of data, where possible.

·

Basic reporting

-

Experimental design

-

Validity of the findings

-

Additional comments

Pdf file attached

---

## Round 0.2 · accepted · Accept

Thank you for addressing the reviewers' comments and for the comprehensive response document provided. This version has been revised to the requirements of the reviewers, and can thus be accepted for publication.

·

Basic reporting

The authors implemented the corrections effectively.

Experimental design

The authors implemented the corrections effectively.

Validity of the findings

The authors implemented the corrections effectively.

Additional comments

The authors implemented the corrections effectively.